# Secretory Phospholipases A2, from Snakebite Envenoming to a Myriad of Inflammation Associated Human Diseases—What Is the Secret of Their Activity?

**DOI:** 10.3390/ijms24021579

**Published:** 2023-01-13

**Authors:** Fiorella Tonello

**Affiliations:** CNR of Italy, Neuroscience Institute, Viale G. Colombo 3, 35131 Padova, Italy; fiorella.tonello@cnr.it

**Keywords:** disulphide modification, phosphorylation, ubiquitination, sumoylation, genetic variants, receptors, phase separation, membrane protein complexes, signal transduction, lipid metabolism

## Abstract

Secreted phospholipases of type A2 (sPLA2s) are proteins of 14–16 kDa present in mammals in different forms and at different body sites. They are involved in lipid transformation processes, and consequently in various immune, inflammatory, and metabolic processes. sPLA2s are also major components of snake venoms, endowed with various toxic and pharmacological properties. The activity of sPLA2s is not limited to the enzymatic one but, through interaction with different types of molecules, they exert other activities that are still little known and explored, both outside and inside the cells, as they can be endocytosed. The aim of this review is to analyze three features of sPLA2s, yet under-explored, knowledge of which could be crucial to understanding the activity of these proteins. The first feature is their disulphide bridge pattern, which has always been considered immutable and necessary for their stability, but which might instead be modulable. The second characteristic is their ability to undergo various post-translational modifications that would control their interaction with other molecules. The third feature is their ability to participate in active molecular condensates both on the surface and within the cell. Finally, the implications of these features in the design of anti-inflammatory drugs are discussed.

## 1. Introduction

A phospholipase A2 (PLA2) is an enzyme that hydrolyzes the second ester bond of a phospholipid releasing a fatty acid and a lysophospholipid. Eucaryotic PLA2s are classified in intra- and extra-cellular or secreted. Secretory PLA2s (sPLA2s) are translated with a N-terminal signal peptide that is removed in the endoplasmic reticulum, and then they are translocated through the Golgi and secretory vesicle to the extracellular space. They are abundant in numerous human body fluids, in pancreatic juice, tears, seminal fluid, and some of them are detectable also in blood serum [1,2].

sPLA2s are also present in prokaryotes, and procaryotic and eukaryotic sPLA2s share the same mechanism of action, in which a water molecule, activated by the histidine residue (H48) of the active site, performs a nucleophilic attack on the sn2 carbonyl oxygen of the phospholipid [3]. The reaction intermediate is stabilized by a calcium ion coordinated by an aspartic acid residue (D49) and the calcium binding loop (GCxCGxG). According to the classification described by Schaloske and Dennis (2006) [4], sPLA2 of groups I, II, V and X, which share the 3D structure and disulphide bonding pattern, belong to a single structural class, later termed ‘conventional’, whereas group III sPLA2, which resemble the sPLA2 of bee venom, and group XII sPLA2, constitute a second and a third structural class and are considered atypical [5].

The best-known sPLA2 functions are the digestion of lipids, in the digestive system, and the antibacterial defense in various body sites, e.g., in the intestines, in tears. However, these proteins are involved in various inflammatory and lipid metabolism processes and consequently in numerous diseases, from metabolic and cardiovascular, to neurodegenerative and neuromuscular diseases. Table 1 shows the main tissues and diseases in which conventional sPLA2s are involved.

Important toxic components of the snake venoms are sPLA2s of group I and II. They are endowed with several toxic properties, among which the most widespread are hemotoxicity, myotoxicity, and neurotoxicity (Table 2). The homology between the phospholipases composing snake venoms and some human sPLA2s is very high, in some cases higher than the homology between human sPLA2s belonging to different subgroups [28]. Therefore, snake venom sPLA2s can be considered a reservoir of natural homologs, selected over millions of years for their toxic properties, which can help us understand the basic function and pathological features of human sPLA2s.

The biological activity of PLA2s, although they are involved in numerous pathologies, has not yet been fully understood. Firstly, because their enzymatic activity, well characterized in vitro, is more difficult to know in vivo because it is influenced by many local parameters, such as the type of lipids present, their accessibility, the micro-local calcium concentration, and interactions with other proteins and carbohydrates. Secondly, because sPLA2s perform other activities besides their enzymatic function. Natural sPLA2 homologs of snake venom lacking catalytic activity, as the aspartic acid residue (D49) in the active site is replaced by lysine (K49) or other amino acids, possess equal or even higher toxicity than their catalytically active counterparts. This non-catalytic activity of sPLA2s is even less known because it depends on a complex network of interactions with many other types of molecules. Finally, the activity of sPLA2s is not only expressed outside the cell because, following interaction with various surface receptors and co-receptors, sPLA2s are internalized [34].

In recent years, many reviews have been focused on these proteins, some addressing one group in particular, group IB [8], including IIA [13,14,16,35,36], II E, D, and F [18], as well as sPLA2s of group V [24] and group X [27], while others relate to the role of sPLA2s in macrophage modulation [37], in insulin resistance and metabolism [38], in acute lung injury [15], as targets of anti-inflammatory drugs [39], in viral diseases [40], and to their non-enzymatic activity [34]. Regarding the non-enzymatic activity of sPLA2s, Ivanušec et al. discussed the interaction of sPLA2s with soluble and membrane proteins, the signaling triggered by the interaction of sPLA2s with cell membrane receptors, and their internalization, intracellular transport, and localization [34]. This review examines three yet underexplored aspects of sPLA2s’ non-enzymatic activity: the role of the disulphide bridge pattern, which may not be as immutable, as has been assumed; the possibility that these proteins, when internalized in the cell, are targets of enzymes that introduce post-translational modifications that contribute to varying and controlling sPLA2 activity; and, third, the protein assemblies in which sPLA2s participate and their possible biological role. Finally, based on these three new features, the development of new drugs targeting sPLA2s in inflammation associated diseases will be discussed.

## 2. sPLA2s Possess a Specific Pattern of Disulphide Bonds That Could Be Modulable

Conventional sPLA2 share a pattern of disulphide, comprising five to eight bridges (Figure 1). The central part of this domain, consisting of four bridges, dates to the basal metazoan. The structural domain shared by these proteins is classified with the code cd00125 in the Conserved Domain Database (CDD, NCBI), and it comprises the functional sites for enzymatic activity: the active site and the calcium binding loop [3].

Most eukaryotic secreted proteins possess disulphide bridges [41], whose best-known function is to confer stability, but which are also often involved in modulating the activity of both secreted proteins and membrane receptors [42], such as in the process of virus internalization [43]. A mutagenesis study of each, one by one, of the seven disulphide bridges of bovine pancreatic sPLA2 (group IB), found that only one (the one that corresponds to number 8 in Figure 1) is crucial for the folding process, while the first bridge (n. 1 in Figure 1) is the most important for the protein stability upon denaturation with guanidinium chloride [44]. The stability of bridge n. 1 has also been confirmed by another group through studies of resistance to proteolysis with trypsin [42]. Only the deletion of bridge n. 3, the one that stabilizes the calcium binding loop, causes a sharp drop in enzyme activity, whereas bridge n. 2, the one in the most disordered part of the protein, is a highly strained chemical bond, and its deletion confers stability to the protein [44]. Intriguingly, bridge n. 2 has been identified as labile in two group IIA sPLAs, the human one and the PLA2 homologue of *Bothrops pirajai*, piratoxin-I, as it is present in some but not all structures of the proteins. This suggests that it can be an allosteric bridge, i.e., a bridge that confers different activities to the protein depending on its state, open or closed [45]. A labile bridge was also identified in a group I sPLA2 of *Naja atra* (Uniprot entry P00598), the one corresponding to number 8 in Figure 1 [45].

A labile disulphide bridge could be the object of attack by cysteines present in other proteins, which would thus form a permanent or temporary covalent bond with sPLA2s, capable of modulating their structure and activity, or involving them in protein complexes. Interestingly, a group II snake venom PLA2, ammodytoxin (Atx), has been reported to have a strong interaction with protein disulphide isomerase (PDI), a chaperone protein that catalyses the opening and closing of disulphide bridges. No covalent link has been found between the two, but it may be a short-lived link, or happen only under certain conditions [46]. Moreover, Atx interacts with cytochrome c oxidase in the mitochondria intermembrane space [47], whose import pathway recognizes proteins that are cysteine-rich, and it is based on a disulphide relay mechanism [48].

The enzymatic activity of human and mouse sPLA2 is inhibited by the presence of dithiothreitol, a reducing agent, but with different efficiency depending on the protein, probably due to a different resistance to the reduction of their disulphide bridges, or due to a persistence of enzymatic activity even after the opening of some bridges [49]. This implies that some of these proteins can also be active in a reducing environment, which is important considering that these proteins are internalized, and that the cytosolic environment is reducing [50]. In the case of Atx, it has been proven that it is also active under cytosolic conditions [51].

In summary, sPLA2s contain a highly conserved pattern of disulphide bridges, not all of which, however, are strictly necessary for the stability and catalytic activity of the protein. Some of these bridges, in particular the second bridge described in Figure 1, may be conserved because they are involved in as yet unknown modulations of the activity of the sPLA2 through interactions with reducing agents or with reduced cysteines of other proteins.

## 3. sPLA2 Activity Can Be Controlled by Post Translational Modifications

sPLA2s are expressed with a N-terminal signal peptide that causes them to be translocated during synthesis into the endoplasmic reticulum, where they acquire the correct three-dimensional structure, and then they are secreted in a conventional manner via secretion vesicles that fuse with the plasma membrane. sPLA2s are not known to undergo post-translational modifications in the ER and Golgi, apart from disulphide bridge closure and signal peptide removal, and in the extra-cellular environment, apart from the removal of a pro-peptide in the case of group I and X PLA2s [1,27]. However, when they are internalized and translocated into the cytosol, they enter in an environment rich in enzymes that can modify them and thus change their activities and molecular interactions.

There are no studies dedicated to the PTMs of sPLA2s, but some have been identified in studies detecting PTMs present in all cellular proteins (Figure 2). Several phosphorylation sites have been identified in group IIE and IID human sPLA2s [52,53]. Group IIA PLA2 was found to be phosphorylated at tyrosine 86 [54], ubiquitinated at lysine 82 [55], and sumoylated at lysine 128 [56]. Sumoylation is a modification that is more frequent in proteins that are transported to the nucleus, and PLA2-IIA can have a nuclear localization.

The possibility that sPLA2s undergo PTM should be considered when studying the molecular effects caused by pathological genetic variants of these proteins [57]. The R143H mutation of PLA2-IIA (R123H considers the numbering without the signal peptide), found in two infants affected by acute respiratory distress syndrome (ARDS) [12], modifies a possible phosphorylation site recognized by cyclin-dependent protein kinase (CDK) ([ST])P[RK]), where a serine or a threonine is followed by a proline at position +1 and a basic residue at position +2 [58]. CDKs are involved in other processes besides cell division, e.g., in the regulation of the immune system [59], so it would be worth investigating whether the pathogenicity of the R143H mutation is due to the alteration of a CDK phosphorylation site. The S80G mutation of PLA2-IID (S80G considers the numbering without the signal peptide), associated with weight loss in subjects with chronic obstructive pulmonary disease (COPD) [19], could alter a phosphorylation site by glycogen synthase kinase-3 (GSK3), a kinase involved in numerous biological processes and diseases [59], which is also a regulator of metabolic pathways [60] involved in muscle atrophy in COPD [61].

A wealth of pathological variants of these proteins are present in animal venoms. By comparing the sequence of sPLA2s from snake venom with the corresponding human homologues, differences emerged in the short linear motifs (SLiMs) conserved in the two protein categories. This comparison showed that human proteins differ from animal venom in the presence of SLiMs susceptible to undergoing PTMs by the S/T kinases CDK, GSK3, PKA, MAPK, by tyrosine kinases, and by Pin1, a [ST] phosphorylation-regulated prolyl isomerase [62]. These differences in the sites subjected to PTMs may give the toxins the possibility of establishing different molecular interactions than mammalian sPLA2s and/or confer different resistance to degradative processes via the ubiquitin-proteasome pathway or micro autophagy.

In conclusion, to investigate the biological action of sPLA2s, it is important to consider that, when they are imported into the cytosol, they may undergo PTMs that affect their molecular interactions and biological activity. The pathological action of human variants of sPLA2s, and of toxic sPLA2s present in animal venoms, could also be due to mutations of their PTM sites.

## 4. sPLA2s Are Globular Proteins, but They Can Form Active Condensates

The sPLA2s interact with different types of proteins, both extra- and intracellular [34]. Basic phospholipases also interact with heparan sulphate proteoglycans (HSPGs) on the cell surface [21,63,64,65,66]. Some sPLA2s can also form catalytically active oligomers when they come into contact with lipid membranes [67], or amyloid-like assemblies on the cell surface, implicated in their internalization [68]. These multiple interactions suggest that sPLA2s are not proteins that establish 1:1 interaction, but rather participate in molecular complexes. Molecular complexes formed by liquid–liquid phase separation (LLPS) are considered membraneless organelles (MLOs) or biomolecular condensates. Several intracellular MLOs are known to perform a variety of functions, from chromatin organization to ribosome assembly, to mRNA transport and metabolism [69].

Two-dimensional condensates can also be formed on the membrane of the cell or internal organelles [70]. Recently, the formation of an extracellular condensate between fibroblast growth factor and HSPG has also been observed, which serves to enrich the growth factor around its receptor and trigger signaling transduction [70]. The same phenomenon can happen with the sPLA2s that interact with HSPGs on the cell surface, and activate an intracellular signal by yet unknown mechanisms, which also triggers their internalization. For example, group II human sPLA2 interacts with glypican-1 (GPC-1), a transmembrane protein modified with heparan sulphates, and thanks to this interaction, is internalised and localised in the paranuclear zone [66]. In addition to HSPGs, two group II sPLA2s, the human one and a PLA2-like myotoxin, interact with two proteins, nucleolin (NCL) and vimentin (VIM), known for their ability to establish multiple molecular interactions both inside the cell, where they are normally found, and on the cell surface, where they are secreted by unconventional mechanisms [36,68]. On the cell surface, NCL and VIM interact with different membrane receptors, forming complexes that mediate the internalization of different types of viruses or other pathogens and proteins [71,72]. Membrane proteins with which NCL and VIM interact include integrins alpha 3, alpha 4, beta 1, and beta 3 (ITGA4, ITGA6, ITGB1, and ITGB3), the epidermal growth factor receptor (EGFR), and glypican-1 (GPC-1), all identified as group II sPLA2 receptors [34] (Figure 3). Group II sPLA2s can be assumed to participate in cell surface molecular complexes involving various membrane proteins, proteoglycans, and secreted factors. Other sPLAs can be involved in these kind of complexes in the cell surface, i.e., Group V human PLA2 was suggested to induce the release of proangiogenic and antiangiogenic factors by interacting with PLA2R1 and heparan sulfate proteoglycans, or with αVβ3 and α4β1 integrins [64].

Because of interaction with the surface receptors described above, sPLA2s are internalized and transported to the paranuclear, or in some cases, even nuclear zones [24,66,74]. The intracellular function of sPLA2s is still poorly understood, but based on the intracellular interactions they establish, it can be assumed that they participate in protein complexes that are formed to produce lipid derivatives, such as prostaglandins and leukotrienes. Many lipid intermediates are highly reactive and have a very short half-life, which is why the enzymes that transform them are assembled into complexes that allow lipid transformation to be completed rapidly and in the same cellular micro zone [75]. Within the cell, sPLA2s colocalise with cytosolic phospholipase, cyclooxygenase 2, and lipoxygenase 12 [66,76,77,78,79]. In addition, several sPLA2 receptors or coreceptors interact with and regulate the assembly of lipid metabolism enzymes: proteoglycans, integrins, and vimentin [76,79,80]. NCL and EFGR, on the other hand, interact with mRNAs of lipid metabolism enzymes and presumably regulate their translation [81,82].

The toxic action of snake sPLA2s is not yet fully understood. At the level of the plasma membrane, in the case of catalytically active phospholipases, one hypothesis is that the enzyme action may destabilize membranes, in the case of myotoxins, or induce the fusion of neurotransmitter vesicles with the synaptic terminal, in the case of neurotoxins [28,83]. In the case of phospholipase-like proteins, one hypothesis is that they can destabilize membranes through interaction with negatively charged lipids, causing an ionic imbalance in the cell. A mechanism analogous to this has also been hypothesized for proteins that form amyloid oligomers, but, in recent years, an alternative hypothesis has emerged: that condensates of these proteins on the plasma membrane may activate harmful intracellular signaling. This alternative hypothesis could also explain the action of snake sPLAs, which activate various signal cascades, thus triggering inflammatory responses and, at high concentrations, even cell death [84,85]. Several venom sPLA2s are internalized and, similar to human sPLA2s, localized in the nuclear or paranuclear zone [68,86,87], where they can alter molecular complexes involved in lipid metabolism and, for instance, cause excess production of reactive oxygen species or toxic lipid derivatives. At the mitochondrial level, Šribar et al. demonstrated that Atx interacts with and inhibits cytochrome c oxidase, a large transmembrane protein complex [47]. Another aspect to consider is the interaction of catalytically inactive sPLA2s with specific lipids. *Bothrops asper* myotoxin-II has a high affinity for phosphatidic acid [83], a lipid that regulates the activity of several integral and peripheral membrane proteins [88] and is also involved in LLPS processes [89].

To summarize, the multiple interactions established by sPLA2s with different types of molecules, lipids, carbohydrates, membrane receptors, co-receptors, and intracellular enzymes mean that they can participate in the formation of molecular condensates both on the cell surface and intracellularly. Understanding how these complexes are formed and broken down, as well as their function, will be crucial to understanding the activity of sPLA2s.

## 5. How to Target sPLA2s in Inflammation-Associated Human Diseases

Most, if not all, diseases in which sPLA2s are involved have an inflammatory component, and A2-type phospholipases, mainly sPLA2s, are considered an important target in the development of anti-inflammatory drugs, as they arbitrate the release of fatty acids that are then converted into inflammatory mediators. Today, inflammation is mainly treated with cyclooxygenase 1 and 2 inhibitors that act downstream of the action of sPLA2s, or with cortisone drugs that act instead upstream of sPLA2s production, as they regulate the expression of different inflammatory agents. Both types of treatments have important side effects, ranging from gastrointestinal and cardiovascular disorders in the case of cyclooxygenase inhibitors, to disorders of various kinds in the case of cortisone drugs, as this class of proteins regulates various physiological processes that are consequently altered [39].

Inhibitors of sPLA2 enzymatic activity have not yielded the desired results in clinical trials [90]. One possible reason is that they are not sufficient to block the enzymatic activity of these proteins. Moreover, their activity is extremely specific, depending on the micro-zone of action, and on the complex of molecular interactions they establish. Knowing the interactions and any molecular condensates in which sPLA2 is involved is of paramount importance. Molecular condensates regulate a variety of cellular activities and are involved in numerous pathological processes [69,91], therefore, they are being studied for the development of molecules that can induce their formation, alter, or even dissolve them [92].

Examples of molecules that interfere with molecular interactions of sPLA2s already exist: an aptamer that binds to NCL, a disordered co-receptor protein of a snake venom sPLA2, preventing its internalization into the cell and protecting it from its toxic effects [68]; and pentapeptides, which mimic a central stretch (63–67 in Figure 2) of Group II sPLA2s, one of which, cyclic, is in clinical trial as a drug against prostate cancer [36,93]. Interestingly, tract 63–67 includes a tyrosine, conserved in groups IIA, IID, and IIE sPLA2s, which has been identified as phosphorylated in sPLA2s IIA and IIE by high throughput investigations (see Section 3 and Figure 2), suggesting that this peptide may act by interfering with this PTM.

In the future, we will see an increase in the development and creation of molecules acting on molecular condensates [94], and we will need a system for screening these molecules also on models of action of sPLA2s. Such a system can be realized by treating cellular cultures with recombinant sPLA2s conjugated to fluorophores [68,86]. In this way, the molecular condensates in which the sPLA2s participate can be visualized, and the effects of the condensate modifiers can be monitored via live imaging.

To sum up, the development of new anti-inflammatory drugs targeting sPLA2s will have to consider the complexity of their action and regulation, as well as the multiple molecular interactions they can establish. To this end, new experimental models will be required to test the action of compounds that affect the formation and stability of the molecular complexes of which they take part.

## Figures and Tables

**Figure 1 ijms-24-01579-f001:**
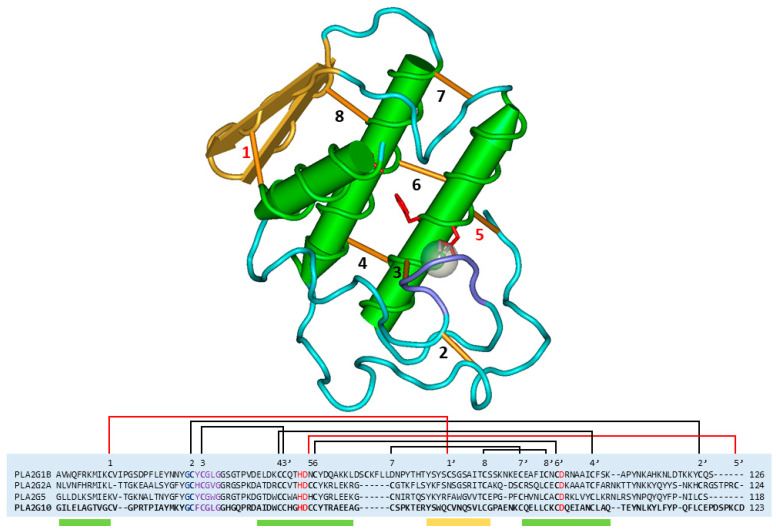
Structure of human sPLA2 of group X and sequence alignment of this with human sPLA2s of the groups IB, IIA, and V. The different sPLA groups are distinguished by the number and position of the disulphide bridges. In the sequence alignment, the conserved bridges in all conventional sPLA2s are shown in black, the active site residues in red, and the calcium binding loop residues in purple. Group X sPLA2s possess eight disulphide bridges, numbered from 1 to 8 according to the order of appearance in the sequence of the first cysteine in the bridge. Group I and group II sPLA2s possess seven bridges, whose cysteines are aligned with those of bridges 1–7 and 2–8 of group X sPLA2, respectively. In contrast, group V sPLA2s have six bridges, whose cysteines are aligned with those of bridges 2–7 of group X sPLA2. The three-dimensional structure representation was obtained from the protein data bank (PDB), file 1LE7, using the Cn3D macromolecular structure viewer.

**Figure 2 ijms-24-01579-f002:**
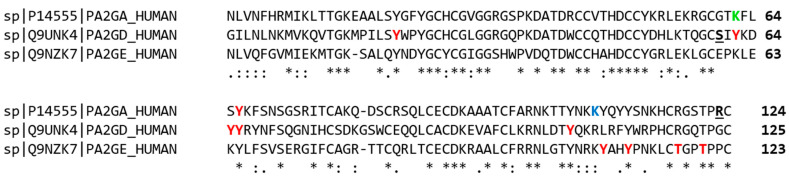
PTM sites and pathological genetic variants of human sPLA2s of group II. High throughput studies of PTMs identified, in these proteins, phosphorylation sites (in red), a ubiquitination site (in green), and a sumoylation site (in blue). Residues mutated in genetic diseases are evidenced in underlined bold. The asterisks indicate fully conserved residues, the colons conservation of amino acids with strong similar properties, while the symbol “.” indicates conservation of amino acids with weakly similar properties.

**Figure 3 ijms-24-01579-f003:**
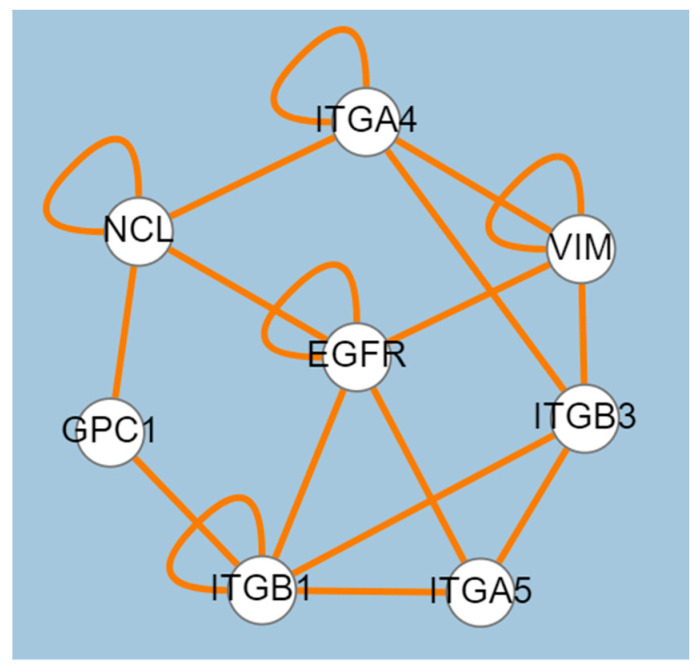
Group II phospholipase receptors or co-receptors have a direct physical interaction with each other, suggesting their involvement in multi-molecular complexes on the cell surface. Figure obtained via EsyN, using high and low throughput data from the Biogrid database [73]. The orange lines represent physical interaction between proteins.

**Table 1 ijms-24-01579-t001:** Expression data and main diseases where conventional sPLA2s are involved.

sPLA2 Group Gene Name	Tissue Specificity *	Single Cell Type Specificity *	Diseases	Refs.
IB *PLA2G1B*	Pancreas, lung, liver, colon, and kidney. At lower levels in ovary and testis, gastric mucosa, spleen, and brain	Exocrine glandular cells, pneumocytes, neuronal cells, plasma cells	Metabolic diseases Behavior disorders Arthritis rheumatoid Spinal cord and brain injury Cancer, mainly of lung, pancreas, liver, pituitary, colon, and adrenal glands	[6,7,8,9]
IIA *PLA2G2A*	Liver, colon and small intestine, eyes, lung, pancreas, placenta, skeletal muscle, prostate, ovary, heart, kidney, synovium	Exocrine glandular cells, adipocytes, epithelium cells, megakaryocyte–erythroid progenitor cells, chondrocytes, oocytes, granulocytes	Gut microbiota modulator Arthritis rheumatoid Atherosclerosis Acute respiratory distress syndrome, Asthma Coronary artery disease Duchenne muscular dystrophy Alzheimer’s disease Spinal cord and brain injury Cancer, mainly of prostate, colon, rectum, liver, breast, and skin	[6,10,11,12,13,14,15,16]
IID *PLA2G2D*	Spleen and lymph nodes, pharynges, pancreas, lacrimal gland. At lower levels in colon, thymus, placenta, small intestine, and prostate	Dendritic cells, macrophages, plasma cells	Psoriasis Infectious diseases Cancer: breast, skin Chronic obstructive pulmonary disease Asthma	[17,18,19]
IIE *PLA2G2E*	Adipose tissue, lacrimal gland, brain, heart, coronary artery, lung, placenta, and hair follicles	Smooth muscle cells, contractile cells, plasma cells, and melanocytes	Metabolic diseases Alopecia Spinal cord injury Ulcerative colitis Chronic rhinosinusitis Cancer: cutaneous melanoma, breast adenocarcinoma	[6,7,18]
IIF *PLA2G2F*	Epidermis, placenta, testis, and thymus. At lower levels in the heart, kidney, liver, and prostate	Keratinocytes, plasma cells, osteoclast, astrocytes, synovial cells, and capillary endothelial cells	Psoriasis Actinic keratosis Arthritis rheumatoid Cancer: myeloma, lymphoma, and skin	[18,20,21]
V *PLA2G5*	Heart, retina, placenta, and adipose tissue. At lower levels in lung	Sertoli cells, Leydig cells, cardiomyocytes, peritubular cells, cone photoreceptor cells, smooth muscle cells, monocytes, and macrophages	Endometriosis Osteoarthritis Acute coronary syndrome Late onset retinal degeneration Familial benign flacked retina Cancer: central nervous system, prostate	[7,22,23,24]
X *PLA2G10*	Spleen, thymus, peripheral blood leukocytes, pancreas, lung, colon, neuronal fibers, white adipose tissue, and prostate	Distal enterocytes, Paneth cells, intestinal goblet cells, exocrine glandular cells, gastric mucus-secreting cells, and pneumocytes	Acute coronary syndrome Cancer: colon-rectum, gastric, and esophagus	[25,26,27]

* Note: sPLA2s expression data are highly variable, as they depend on the condition, and state of activity and health, of the subject.

**Table 2 ijms-24-01579-t002:** Main toxic effects caused by animal venom PLA2s, and number of proteins having that toxic property. Data were obtained through an advanced search in UniProtKB, selecting the protein family (phospholipase A2), the taxonomy (serpentes), and the keyword corresponding to the type of toxicity.

Toxic Effect	Number of Proteins	Description	Refs.
Edema	Group I: 4 Group II: 82	Local swelling that may appear within 15 min, spread, and become massive in two to three days and persist for up to three weeks	[29]
Hemostasis impairing	Group I: 35 Group II: 66	Many venom PLA2s have anticoagulant activity, some by inhibition of coagulation factors (FXa, Factor VII), others by interfering with platelet aggregation. Platelet aggregation can be affected by reaction product of the phospholipase activity, however, not all PLA2s affect platelet aggregation, despite their common catalytic activity. Another possibility is the alteration of membrane receptors.	[30,31]
Myotoxins	Group I: 11 Group II: 81 ^1^	The myotoxic activity of PLA2s can be local or systemic. Some myotoxins act only on muscle cells, and others have less specific activity and may be termed cytotoxins.	[32]
Neurotoxins	Group I: 38 ^2^ Group II: 53 ^3^	Venom neurotoxic PLA2s mostly act at the pre-synaptic level. Their specificity is thought to be due to interaction with membrane receptors. They cause paralysis by interfering with the release of acetyl choline.	[33]

^1^ 53 of 81 group II myotoxic PLA2 are enzymatically inactive homologues; ^2^ 14 of 38 group I neurotoxic PLA2 are part of heterodimers or oligomers; ^3^ 21 of 53 group II neurotoxic PLA2 are part of heterodimers or oligomers.

## Data Availability

Not applicable.

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
