# Peer review of "Secretory Phospholipases A2, from Snakebite Envenoming to a Myriad of Inflammation Associated Human Diseases—What Is the Secret of Their Activity?"

_ijms, 2023, doi:10.3390/ijms24021579_

Round 1
Reviewer 1 Report
Dr. Fiorella describes three characteristics of Secreted phospholipases of type A2 that can help us understand the biological activities of these kinds of proteins. To improve the manuscript, I recommend taking into account the following observations:
The sentence in lines 32, 33, and 34 needs a supporting citation. Citation 26, in line 51, does not support the author's writing.
I suggest the author write "No. protein" instead "No." in the second column of table No. 2.
I suggest replacing the word "mutants" with "homologs" when referring to the change of D49 to K49 from lines 52 and 63.
I recommend a better connection in the paragraph (lines 76-81) with the previous one so there is a smoother reading at the end of this section.
The section "Introduction" is based on the order of the phospholipase presentation based on the classification described by Schaloske and Dennis (2006) (reference 3). Therefore, it is necessary to mention their classification more explicitly in the document's text between the first and second paragraphs.
Line 114 refers to the species Naja atra and is written as Naia atra. Therefore, it is necessary to modify it.
In the paragraph between lines 126-129, supported by the citation [49], it should be mentioned that the statement corresponds to PLA2 from humans and mouse and not to all secretory proteins.
Author Response
I thank the reviewer for his careful reading of the manuscript and for his comments to which I provide a point-by-point response below.
The sentence in lines 32, 33, and 34 needs a supporting citation. Citation 26, in line 51, does not support the author's writing.
I inserted the citation, which I had forgotten to provide, regarding the sentences in lines 32, 33,and 34. Regarding the sentence in line 51 I specified that the homology between snake venom and human sPLA2s is 'in some cases higher’ than the homology between human sPLA2s belonging to different subgroups. The cited review (now citation number 28) contains a figure with sequence identity between different human and snake venom PLA2s.
I suggest the author write "No. protein" instead "No." in the second column of table No. 2.
Thanks for the suggestion, I changed the header of the second column of table 2
I suggest replacing the word "mutants" with "homologs"when referring to the change of D49 to K49 from lines 52 and 63.
Thanks for the suggestion, I replaced the word 'mutants' with 'homologs'.
I recommend a better connection in the paragraph (lines 76-81) with the previous one so there is a smoother reading at the end of this section.
I have added a connecting sentence between the two paragraphs specifying that this review completes the analysis on the non-enzymatic properties of sPLA2s.
The section "Introduction" is based on the order of the phospholipase presentation based on the classification described by Schaloske and Dennis (2006) (reference 3). Therefore, it is necessary to mention their classification more explicitly in the document's text between the first and second paragraphs.
Right observation, the classification should have been commented more explicitly, now I have replaced the sentence with this: ‘According to the classification described by Schaloske and Dennis (2006) [4], sPLA2 of groups I, II, V and X, which share the 3D structure and disulphide bonding pattern, belong to a single structural class, later termed 'conventional', whereas group III sPLA2, which resemble the sPLA2 of bee venom, and group XII sPLA2, constitute a second and a third structural class and are considered atypical [5].
Line 114 refers to the species Naja atra and is written as Naia atra. Therefore, it is necessary to modify it.
I have corrected the spelling mistake.
In the paragraph between lines 126-129, supported by the citation [49], it should be mentioned that the statement corresponds to PLA2 from humans and mouse and not to all secretory proteins.
I specified that the statement is about human and mouse PLA2s.
Reviewer 2 Report
PLA2s represent a big protein family which widely distribute in mammals and animal venoms. Although there were numerous studies on PLA2s structure and functions, the underlying mechanism remains mysterious. This manuscript tried to interpret the secret of PLA2’s activities concerning to their disulphide bridge pattern, PTMs and PPI. These three aspects are common mechanisms related to protein activities, so it is undoubtedly reasonable in explanation of the myriad of PLA2s activities. However, I suggest the author to improve the manuscript with the following points
1. Table 2, it’s would be better to list out the categories and structural characteristics of PLA2s. The number from Uniprot database remains obscure to reader, what’s more, the literatures (ref 27-31) in table 2 were old.
2. I suggest to add a footnote below table 1 to list out the protein name of PLA2s corresponding to their gene name
Author Response
I would like to thank the reviewer for his comments and suggestions that allowed me to improve the article and to which I will respond point by point below.
Table 2, it’s would be better to list out the categories and structural characteristics of PLA2s. The number from Uniprot database remains obscure to reader, what’s more, the literatures (ref 27-31) in table 2 were old.
I modified the table by specifying the number of PLA2s, with the different types of toxicity, belonging to group I and to group II. I specified in the table caption how I obtained the numbers reported in the table and added notes to indicate the number of proteins that are inactive homologues and the number of proteins that are part of oligomers. I have also modified the bibliography by adding more recent citations.
I suggest to add a footnote below table 1 to list out the protein name of PLA2s corresponding to their gene name
Instead of adding a footnote, I added the PLA2s protein groups directly in the column, in addition to the gene names.